# Mitochondrial Calcium-Triggered Oxidative Stress and Developmental Defects in Dopaminergic Neurons Differentiated from Deciduous Teeth-Derived Dental Pulp Stem Cells with MFF Insufficiency

**DOI:** 10.3390/antiox11071361

**Published:** 2022-07-13

**Authors:** Xiao Sun, Shuangshan Dong, Hiroki Kato, Jun Kong, Yosuke Ito, Yuta Hirofuji, Hiroshi Sato, Takahiro A. Kato, Yasunari Sakai, Shouichi Ohga, Satoshi Fukumoto, Keiji Masuda

**Affiliations:** 1Section of Oral Medicine for Children, Division of Oral Health, Growth and Development, Faculty of Dental Science, Kyushu University, Maidashi 3-1-1, Higashi-Ku, Fukuoka 812-8582, Japan; sunxiao1988@dent.kyushu-u.ac.jp (X.S.); tousousan@dent.kyushu-u.ac.jp (S.D.); kamea123123@dent.kyushu-u.ac.jp (J.K.); y-ito.1010@dent.kyushu-u.ac.jp (Y.I.); hirofuji@dent.kyushu-u.ac.jp (Y.H.); hisato@dent.kyushu-u.ac.jp (H.S.); 2Department of Molecular Cell Biology and Oral Anatomy, Graduate School of Dental Science, Kyushu University, Maidashi 3-1-1, Higashi-Ku, Fukuoka 812-8582, Japan; kato@dent.kyushu-u.ac.jp; 3Department of Neuropsychiatry, Graduate School of Medical Sciences, Kyushu University, Maidashi 3-1-1, Higashi-Ku, Fukuoka 812-8582, Japan; takahiro@npsych.med.kyushu-u.ac.jp; 4Department of Pediatrics, Graduate School of Medical Sciences, Kyushu University, Maidashi 3-1-1, Higashi-Ku, Fukuoka 812-8582, Japan; ysakai22q13@gmail.com (Y.S.); ohgas@pediatr.med.kyushu-u.ac.jp (S.O.)

**Keywords:** EMPF2, mitochondrial calcium, mitochondrial fission factor, reactive oxygen species, stem cells from human exfoliated deciduous teeth

## Abstract

Mitochondrial fission factor (MFF) is an adapter that targets dynamin-related protein 1 from the cytosol to the mitochondria for fission. Loss-of-function MFF mutations cause encephalopathy due to defective mitochondrial and peroxisomal fission 2 (EMPF2). To elucidate the molecular mechanisms that were involved, we analyzed the functional effects of MFF depletion in deciduous teeth-derived dental pulp stem cells differentiating into dopaminergic neurons (DNs). When treated with MFF-targeting small interfering RNA, DNs showed impaired neurite outgrowth and reduced mitochondrial signals in neurites harboring elongated mitochondria. MFF silencing also caused mitochondrial Ca^2+^ accumulation through accelerated Ca^2+^ influx from the endoplasmic reticulum (ER) via the inositol 1,4,5-trisphosphate receptor. Mitochondrial Ca^2+^ overload led DNs to produce excessive reactive oxygen species (ROS), and downregulated peroxisome proliferator-activated receptor-gamma co-activator-1 alpha (PGC-1α). MFF was co-immunoprecipitated with voltage-dependent anion channel 1, an essential component of the ER-mitochondrial Ca^2+^ transport system. Folic acid supplementation normalized ROS levels, PGC-1α mediated mitochondrial biogenesis, and neurite outgrowth in MFF depleted DNs, without affecting their mitochondrial morphology or Ca^2+^ levels. We propose that MFF negatively regulates the mitochondrial Ca^2+^ influx from the ER. MFF-insufficiency recapitulated the EMPF2 neuropathology with increased oxidative stress and suppressed mitochondrial biogenesis. ROS and mitochondrial biogenesis might be potential therapeutic targets for EMPF2.

## 1. Introduction

Mitochondrial fission and fusion must be balanced to maintain proper morphology and function, including oxidative phosphorylation, redox control, and calcium regulation [1,2,3]. An imbalance between mitochondrial fission and fusion leads to excessively fragmented or tubular mitochondrial morphology, which is associated with pathological conditions [4]. Several mitochondrial encephalopathies are caused by loss-of-function mutations in genes encoding proteins that are involved in mitochondrial fission and fusion [5].

Encephalopathy due to defective mitochondrial and peroxisomal fission 1 (EMPF1; MIM # 614388) is caused by loss-of-function mutations in dynamin-related protein 1 (DRP1) and is mainly characterized by severe developmental delay and hypotonia. DRP1 is an essential protein that executes both mitochondrial and peroxisomal fission [6,7,8,9,10,11,12,13]. Fibroblasts that were derived from patients with EMPF1 showed an abnormally expanded morphology in both organelles [7,8,12]. We have reported on conditional Drp1-deficient mice by using the Cre-loxP system to elucidate neuropathological mechanisms that are associated with Drp1-deficiency [14]. Mice with a neuron-specific Drp1 deficiency died shortly after birth due to marked hypoplasia of the brain [14]. Primary cultured neurons that were derived from the mutant mice showed an uneven mitochondrial distribution that aggregated mainly in the cell body [14]. This suggests that neurite development and synaptic formation require DRP1-dependent mitochondrial fission to generate compact mitochondria and distribute them in developing neurites and the synaptic terminus. Thus, the failure of mitochondrial distribution due to Drp1 deficiency is considered to be a central mechanism of the neurodevelopmental defects that are associated with EMPF1.

Recently, loss-of-function mutations in mitochondrial fission factor (MFF) were observed to cause EMPF2 (MIM # 617086). MFF is a C-tail-anchored protein that is constitutively located in the mitochondria and peroxisomes [15]. While DRP1 is recruited from the cytosol to the mitochondria and peroxisomes to execute their fission [16,17], in mitochondria, MFF is one of the adaptors targeting DRP1 for fission [2,5,18]. All currently identified loss-of-function mutations of MFF generate a truncated cytoplasmic N-terminal region lacking the tail anchor, indicating impaired mitochondrial and peroxisomal localization [19,20,21,22]. Because MFF is a critical component of the DRP1-dependent mitochondrial fission machinery, the underlying neuropathology of EMPF2 is presumed to be a dysregulation of mitochondrial morphology that overlaps with EMPF1. However, the exact mechanisms of EMPF2 are not fully understood.

The current study aims to elucidate the neuropathological mechanisms of EMPF2, with a particular focus on the developmental defects of dopaminergic neurons (DNs). Previous studies suggested that an enhanced mitochondrial Ca^2+^ buffering capacity led to cortical neuron-specific defects in EMPF2 [23], and mitochondrial Ca^2+^ accumulation harmed the basal ganglia [3,24,25]. We therefore rationalized the utility of stem cells from human exfoliated deciduous teeth (SHEDs) and their differentiation potential into DNs as a disease model of EMPF2 [26,27,28,29,30,31]. We herein report that MFF insufficiency causes mitochondrial Ca^2+^ accumulation and oxidative stress, which may contribute to the neuropathological mechanisms of EMPF2.

## 2. Materials and Methods

### 2.1. SHED Isolation, Culture, and Differentiation into Dopaminergic Neurons

Experiments using human samples were reviewed and approved by the Kyushu University Institutional Review Board for Human Genome/Gene Research (permission number: 678-03), and conducted as per the Declaration of Helsinki. Written informed consent was obtained from the parents. Deciduous teeth were collected from a typically developing six-year-old boy.

SHED isolation was performed as previously described [32]. SHEDs were cultured in the alpha modification of Eagle’s Medium (Nacalai Tesque, Kyoto, Japan) with 15% fetal bovine serum (Sigma-Aldrich, St. Louis, MO, USA); 100 μM of L-ascorbic acid 2-phosphate (Wako Pure Chemical Industries, Osaka, Japan); 250 μg/mL of fungizone (Life Technologies, Tarrytown, NY, USA); 100 U/mL of penicillin; and 100 μg/mL of streptomycin (Nacalai Tesque) at 37 °C in an atmosphere containing 5% CO_2_.

Differentiation from SHEDs to DNs was performed using the two-step procedure that was described by Fujii et al. [28]. In the first step, 1.5 × 10^5^ SHEDs were plated in a six-well culture plate (Corning, NY, USA) and cultured in the culture medium. After 24 h, the cells were cultured in the first-step medium, Dulbecco’s Modified Eagle’s Medium (#08456-65; DMEM, Nacalai Tesque), supplemented with 20 ng/mL of epidermal growth factor (PeproTech, East Windsor, NJ, USA); 20 ng/mL of basic fibroblast growth factor (Peprotech); and 1% N_2_ supplement (Thermo Fisher Scientific, Waltham, MA, USA) for 2 d at 37 °C in an incubator with 5% CO_2_. In the second step, the first-step medium was replaced by a neurobasal medium (Thermo Fisher Scientific), and the second-step medium was supplemented with 2% B27 supplement (Thermo Fisher Scientific, Waltham, MA, USA); 1 mM of dibutyryladenosine 3,5-cyclic monophosphate (Nacalai Tesque, Kyoto, Japan); 0.5 mM of 3-isobutyl-1-methyl-xanthine (Wako Pure Chemical Industries); and 200 μM of ascorbic acid (Nacalai Tesque, Kyoto, Japan). The cells were incubated for 5 d at 37 °C in an incubator with 5% CO_2_.

### 2.2. RNA Interference to Knock down MFF Expression

After the first step of DN differentiation, the first-step medium was replaced by the second-step medium and small interfering RNA (siRNA) transfection was performed with Lipofectamine RNAiMAX (Thermo Fisher Scientific, Waltham, MA, USA). The MFF siRNA sequences were as follows: sense 5′-AACGCUGACCUGGAACAAGGATT-3′ and antisense 5′-UCCUUGUUCCAGGUCAGCGUUTT-3′. The control siRNA was purchased from Sigma-Aldrich, Burlington, MA, USA, SIC001-10NMOL.

### 2.3. Folic Acid, Ruthenium Red, and Xestospongin C Treatment

Folic acid (FA; 20 μM; Wako Pure Chemical Industries) and ruthenium red (Ru-R; 1 μM; Wako Pure Chemical Industries) were added in the second-step medium of DN differentiation. Xestospongin C (Xest-C; 5 μM; Cayman Chemical Company, Ann Arbor, MI, USA) was added after the second step of DN differentiation and the cells were incubated for 4 h at 37 °C in an incubator with 5% CO_2_.

### 2.4. Immunocytochemistry

The DNs were cultured on the cover glass and fixed with 4% paraformaldehyde in 0.1 M of sodium phosphate buffer (pH 7.4) for 10 min at room temperature, then permeabilized with 0.1% Triton X-100 in phosphate-buffered saline (PBS) for 5 min at room temperature. The cells were blocked with 2% BSA in PBS for 20 min at room temperature and then incubated with the following primary antibodies for 90 min at room temperature: mouse monoclonal anti-Tom20 (#sc-17764; Santa Cruz Biotechnology, Paso Robles, CA, USA); mouse monoclonal anti-tyrosine hydroxylase (TH; #66334-1-Ig; Proteintech, Rosemont, IL, USA); mouse monoclonal anti-β-tubulin III (#T8578; Sigma-Aldrich, Burlington, MA, USA); and rabbit polyclonal anti-dopamine (#ab6427; Abcam, Cambridge, UK) antibodies. The cells were subsequently incubated with Alexa Fluor-conjugated secondary antibodies (Thermo Fisher Scientific, Waltham, MA, USA) for 1 h at room temperature in the dark. The information of the Alexa Fluor-conjugated secondary antibodies that are used in this study is presented in Appendix A. After staining with secondary antibodies, the nuclei were counterstained with 1 µg/mL of 4′,6-diamidino-2-phenylindole dihydrochloride (DAPI; Dojindo, Kumamoto, Japan) in PBS for 5 min at room temperature. The cover glass was then mounted on slides using ProLong Diamond mounting medium (Thermo Fisher Scientific, Waltham, MA, USA). Fluorescence images were acquired using CFI Plan Apochromat Lambda 20×, 60×, and 100× objective on a Nikon C2 confocal microscope (Nikon, Tokyo, Japan).

### 2.5. Analyses of Neuronal Morphology, Mitochondrial Amount, Distribution, and Length in Dopaminergic Neurons

Analyses of neuronal morphology and mitochondrial volume in the DNs were performed as previously described [33]. To measure the maximum neurite length and the total number of neurite branches, TH- and DAPI-stained pictures were acquired and 30 cells of each case were analyzed with the Neurite Outgrowth module in MetaMorph software version 7.8 (Molecular Devices, San Jose, CA, USA).

To evaluate the mitochondrial amount in the DNs, pictures of immunofluorescence staining for Tom20 (in the mitochondrial area) and TH (in the cell area) were acquired and analyzed for 30 cells of each case using the Multi Wavelengths Cell Scoring module in the MetaMorph software. The Tom20-stained area was divided by the TH-stained area to determine the total mitochondrial amount in the DNs of each case.

To evaluate the mitochondrial distribution in the neurites, the number of neurites with at least one Tom20-stained area and the total neurites were manually counted for 30 randomly selected cells from the fluorescence images in each case. Thereafter, the number of neurites with at least one Tom20-stained area was divided by the total number of neurites to determine the proportion of mitochondria-containing neurites.

To evaluate mitochondrial length in neurites, 10 mitochondria in the neurites were randomly selected from Tom20- and TH-stained fluorescence images. The mitochondrial length was measured using the ImageJ software version 1.53 [34].

### 2.6. Western Blotting

The DNs were lysed with sodium dodecyl sulfate (SDS) sample buffer containing 62.5 mM of Tris-HCl buffer (pH 6.8), 2% SDS, 5% β-mercaptoethanol, and 10% glycerol, and incubated for 5 min at 95 °C. The proteins in the cell lysates were electrophoresed using SDS-polyacrylamide gel electrophoresis; further, immunoblotting was performed using the rabbit polyclonal anti-peroxisome proliferator-activated receptor gamma coactivator 1 alpha (PGC-1α; #NBP1-04676; Novus Biologicals, Littleton, CO, USA); rabbit polyclonal anti-MFF (#17090-1-AP; Proteintech, Rosemont, IL, USA); rabbit polyclonal anti-voltage dependent anion channel 1 (VDAC1; #55259-1-AP, Proteintech, Rosemont, IL, USA); mouse monoclonal anti-DRP1 (#611113; BD Biosciences, San Jose, CA, USA); mouse monoclonal anti-α-tubulin (#sc-32293; Santa Cruz Biotechnology, Paso Robles, CA, USA); mouse monoclonal anti-β-actin (#66009-1-Ig; Proteintech, Rosemont, IL, USA); horseradish peroxidase (HRP)-linked goat polyclonal anti-mouse IgG (#7076S; Cell Signaling Technology, Danvers, MA, USA); HRP-linked goat polyclonal anti-rabbit IgG (#7074S; Cell Signaling Technology) antibodies; and TidyBlot (for detecting the immunoprecipitants; #STAR209P; Bio-Rad, Hercules, CA, USA).

The immunoreactive bands were detected using ECL Prime (Cytiva, Marlborough, MA, USA) and analyzed using the LAS-1000 Pro (Fuji Film, Tokyo, Japan) and Image Gauge software version 2.11 (Fuji Film). The signals of the objective proteins were normalized to that of α-tubulin signals.

### 2.7. Immunoprecipitation

The DNs were washed with PBS and collected by centrifugation at 800× *g* for 5 min at 4 °C. The cell pellets were lysed in 500 µL of ice-cold immunoprecipitation (IP) lysis buffer, 150 mM of NaCl; 50 mM of Tris-HCl (pH 7.5); 1% CHAPS (Dojindo); 1 mM of EDTA; 1% protease inhibitors (Nacalai Tesque, Kyoto, Japan); and PhosSTOP (Sigma-Aldrich Burlington, MA, USA). Then, the samples were centrifuged at 15,000× *g* for 30 min at 4 °C. The supernatant was collected and incubated with 1 μg of rabbit IgG (#X090302; Agilent, Santa Clara, CA, USA); rabbit polyclonal anti-MFF (#17090-1-AP; Proteintech, Rosemont, IL, USA); or rabbit polyclonal anti-VDAC1 (#55259-1-AP; Proteintech, Rosemont, IL, USA) antibodies overnight at 4 °C. The reaction mixture was incubated with protein A Mag Sepharose Xtra (Cytiva, Marlborough, MA, USA) for 1 h at 4 °C with rotation. Protein A Mag Sepharose Xtra was washed thrice with 500 µL of ice-cold IP lysis buffer and the immunoprecipitants were eluted by SDS sample buffer.

### 2.8. Quantitative Reverse Transcription Polymerase Chain Reaction

Total RNA extraction and quantitative reverse transcription polymerase chain reaction (RT-qPCR) were performed as previously described [33]. The sequence information of the primer sets that are used in this study are listed in Table 1. The relative expression of the target gene was analyzed using the comparative threshold cycle method by normalizing to 18S rRNA expression.

### 2.9. Analyses of Intracellular, Mitochondrial, and Endoplasmic Reticulum Ca^2+^ Levels

The intracellular and mitochondrial Ca^2+^ levels were measured as described previously [35]. The Ca^2+^ levels in the endoplasmic reticulum (ER) were measured using Mag-Fluo-4 AM (AAT Bioquest, Sunnyvale, CA, USA). The cells were incubated with 1 μM of Fluo-4 AM (to detect intracellular Ca^2+^; Thermo Fisher Scientific) for 1 h; 10 μM of Rhod-2 AM (to detect mitochondrial Ca^2+^; Dojindo) for 45 min; or 1 μM of Mag-Fluo-4 AM (to detect ER Ca^2+^) for 45 min. After staining with these probes, the fluorescence images were taken using a CFI Plan Apochromat Lambda 60× objective lens on a Nikon C2 confocal microscope. The fluorescence signals were measured using the EnSight plate reader (PerkinElmer, Hopkinton, MA, USA).

### 2.10. Analyses of NADH and NAD^+^ Levels

The SHEDs (1.5 × 10^5^) were seeded in six-well plates (Corning) and differentiated into DNs. The NADH and NAD^+^ levels were measured using an NAD/NADH assay kit (Dojindo), according to the manufacturer’s manual. Absorbance was measured at 450 nm using the SpectraMax iD3 Microplate Reader (Molecular Devices).

### 2.11. Analysis of Dopamine Levels

The intra- and extracellular dopamine levels in the DNs were evaluated as described previously [35]. To measure the intracellular dopamine levels, the images of dopamine and TH (in the cell area) staining were acquired and 30 cells of each control- and MFF-siRNA group were analyzed with the Multi Wavelengths Cell Scoring module in MetaMorph software version 7.8. The dopamine signal intensity was divided by the TH-stained surface area. After stimulation with 50 mM of KCl, the DNs were immediately fixed with 4% PFA and stained with anti-dopamine antibodies, and the dopamine levels were quantified as described above.

Extracellular dopamine was measured using a Dopamine ELISA Kit (Elabscience, Houston, TX, USA), according to the manufacturer’s instructions. The culture medium (50 μL) was collected to measure extracellular dopamine. To measure extracellular dopamine under KCl stimulated conditions, the cells were treated with 50 mM of KCl for 1 min at 37 °C before harvesting the medium.

### 2.12. Measurement of Mitochondrial Reactive Oxygen Species

Mitochondrial reactive oxygen species (ROS) levels were determined using flow cytometry and confocal microscopy, as previously described [33]. To determine mitochondrial ROS levels using flow cytometry, the cells were incubated with 5 µM of MitoSOX Red (Thermo Fisher Scientific) for 30 min. The cells were subsequently treated with TrypLE Express (Thermo Fisher Scientific) to detach them from the culture plate. The fluorescence signal of 10,000 cells was measured using a FACSCalibur instrument (BD Biosciences). The geometric means of the fluorescence signals were measured using the Cell Quest software version 3.3 (BD Biosciences).

To determine the mitochondrial ROS levels using confocal microscopy, the cells were cultured in μ-dishes (Ibidi, Munich, Germany) and subsequently incubated with 5 µM of MitoSOX Red (Thermo Fisher Scientific) and 20 nM of MitoTracker Green FM (MTG) (Thermo Fisher Scientific) for 30 min. Fluorescence images of MitoSOX Red and MTG were acquired using a Nikon C2 confocal microscope. The fluorescence intensity of MitoSOX Red and MTG was measured using the NIS-Elements AR software version 4.00.06 64-bit (Nikon).

### 2.13. Measurement of Mitochondrial Membrane Potential

The SHEDs (1.5 × 10^5^) were plated in a six-well culture plate (Corning). After differentiation into DNs, the mitochondrial membrane potential (MMP) was measured using the MMP indicator JC-1 (Wako Pure Chemical Industries). The DNs were incubated with 1 µM JC-1 for 10 min. The cells were then treated with TrypLE Express to detach them from the culture plates and JC-1 red and green signals were measured using a FACSCalibur instrument. The geometric means of the red and green fluorescence were measured using Cell Quest software version 3.3, and the ratio of red/green fluorescence was calculated.

### 2.14. Analysis of Intracellular Adenosine Triphosphate Levels

Intracellular adenosine triphosphate (ATP) levels were measured, as previously described [33]. The cells were harvested in ice-cold PBS, and the CellTiter-Glo Luminescent Cell Viability Assay (Promega, Fitchburg, WI, USA) was then used to measure the intracellular ATP levels.

### 2.15. Statistical Analyses

Statistical analyses were performed using Student’s *t*-tests with Prism9 (GraphPad, San Diego, CA, USA). Values are presented as the mean ± standard error of the mean (SEM). *p* < 0.05 indicated statistical significance.

## 3. Results

### 3.1. Impaired Mitochondrial Morphology and Neurite Development in Dopaminergic Neurons with MFF Insufficiency

To investigate the role of MFF insufficiency on DN development and function, cells were treated with siRNA to silence MFF (MFF-siR) or negative control siRNA (Ctrl-siR) during differentiation. Low levels of MFF signals were detected only by the MFF-siR when compared to Ctrl-siR in the immunofluorescence images (Appendix A). A quantitative analysis showed that the MFF expression in the DNs was reduced to approximately 20% in the MFF-siR group, as compared to that in the Ctrl-siR group (Appendix A). The levels of DRP1 and the mitochondrial dynamics proteins of 49 and 51 kDa (MID49 and MID51, respectively), both of which are other DRP1-adaptors on the mitochondrial outer membrane (MOM), were not severely affected by the MFF-siR (Appendix A). Mitochondrial morphology was examined using immunofluorescence staining of Tom20, a marker of mitochondria, and TH, a marker of DNs. The average mitochondrial length was longer, and the Tom20-stained area per cell area and number of mitochondria-containing neurites were reduced in the MFF-siR group compared to the Ctrl-siR group (Figure 1a,b). An immunofluorescence analysis of the neuron morphology showed a shorter maximum neurite length and fewer neurite branches in the MFF-siR group (Figure 1c). The mRNA expression levels of NURR1, a transcription factor that is essential for DN differentiation, and its downstream target TH, were not significantly altered by the MFF-siR (Appendix A). These results suggested that MFF insufficiency caused mitochondrial elongation, reduced the mitochondrial amount and distribution, and impaired neurite development.

### 3.2. Mitochondrial Ca^2+^ and ROS Accumulation in Dopaminergic Neurons with MFF Insufficiency

To clarify the functional alteration of mitochondria that appeared to be elongated in the MFF-siR group, Ca^2+^ levels were examined. Mitochondrial Ca^2+^ levels, rather than the cellular Ca^2+^ levels, were increased in the MFF-siR group compared to the Ctrl-siR group, which was also observed in immunofluorescence images that focused on neurites (Figure 2a and Appendix A). The physiological elevation of mitochondrial Ca^2+^ levels stimulates the tricarboxylic acid (TCA) cycle and oxidative phosphorylation [3,24,25]. Ca^2+^ overload can also induce excessive ROS generation, leading to severe mitochondrial and cellular damage [3,24,25]. To assess the effect of increased mitochondrial Ca^2+^ levels that are caused by the MFF-siR, the levels of NADH, which is generated via the TCA cycle, were measured. Consistent with the increased mitochondrial Ca^2+^ levels, the levels of NADH, rather than NAD^+^, were increased in the MFF-siR group (Figure 2b). Mitochondrial ROS levels were increased by the MFF-siR, which was also observed in the confocal microscopy images that focused on neurites (Figure 2c and Appendix A). However, MFF-siR did not induce apoptotic cell death (Appendix A). Next, we examined whether mitochondrial Ca^2+^ and ROS accumulation also occurred in mitochondria that were elongated by the DRP1 insufficiency. DRP1 inhibition by siRNA did not alter MFF expression levels (Appendix A). This inhibition resulted in mitochondrial elongation but not Ca^2+^ accumulation (Appendix A). However, as shown in a previous report [36], mitochondrial ROS levels were elevated by DRP1 inhibition (Appendix A). These results suggest that MFF insufficiency might cause mitochondrial Ca^2+^ accumulation in the matrix to stimulate the TCA cycle and overproduction of ROS.

### 3.3. Suppression of Mitochondrial Ca^2+^ and ROS Levels by Blocking the Mitochondrial Calcium Uniporter Channel in Dopaminergic Neurons with MFF Insufficiency

The mitochondrial calcium uniporter channel (mtCU) is located in the mitochondrial inner membrane (MIM) and is one of the critical regulators of Ca^2+^ entry into the mitochondrial matrix [3,37,38]. To elucidate the mechanisms of mitochondrial Ca^2+^ accumulation that is caused by MFF insufficiency, mtCU was blocked by its specific reagent, ruthenium red (Ru-R) [39]. Ru-R was added in the second step of DN differentiation, and the cells were subsequently cultured for 5 days (Figure 3). Upon Ru-R treatment, the elevated mitochondrial Ca^2+^ levels in the MFF-siR group were reduced to levels that were comparable to the untreated Ctrl-siR group (Figure 3a), while cytosolic Ca^2+^ levels were increased in the MFF-siR group that was treated with Ru-R (Figure 3b). Thus, in the MFF-siR group, mtCU inhibition might promote Ca^2+^ retention outside mitochondria or Ca^2+^ efflux from mitochondria, resulting in increased cytosolic Ca^2+^ levels. To confirm the active Ca^2+^ efflux from the mitochondria, Ru-R was applied only for a short period (4 h). This treatment also reduced the mitochondrial Ca^2+^ levels in the MFF-siR group to levels similar to that of the Ctrl-siR group, suggesting that mitochondrial Ca^2+^ was promptly released by mtCU inhibition (Appendix A). Furthermore, Ru-R treatment suppressed the mitochondrial ROS levels in the MFF-siR group to levels comparable to those seen in the untreated Ctrl-siR group (Figure 3c). This was also observed in the short-time treatment of Ru-R (Appendix A). Thus, mitochondrial Ca^2+^ accumulation that is caused by MFF insufficiency might be predominantly caused by excessive Ca^2+^ influx through mtCU, which might trigger ROS generation.

### 3.4. Suppression of Mitochondrial Ca^2+^ and ROS Levels by Blocking IP3R in Dopaminergic Neurons with MFF Insufficiency

Apart from the mitochondria, the ER is also a major intracellular Ca^2+^ storage compartment, and Ca^2+^ moves from the ER to the mitochondria. The inositol 1,4,5-trisphosphate receptor (IP3R) of the ER membrane is a regulatory component of this pathway [40]. To clarify the role of the ER in mitochondrial Ca^2+^ accumulation that is caused by MFF insufficiency, we used Xest-C, a specific inhibitor of IP3R [41]. The elevated mitochondrial Ca^2+^ levels in the MFF-siR group were reduced by Xest-C treatment to levels that were comparable to the untreated Ctrl-siR group (Figure 4a). Conversely, the Ca^2+^ levels of the ER were increased in the MFF-siR group that was treated with Xest-C (Figure 4b). Thus, in the DNs with MFF insufficiency, mitochondrial Ca^2+^ accumulation might be caused by enhanced Ca^2+^ influx from the ER through the activation of IP3R. Together with IP3R, the VDAC1 of MOM is an important component of the Ca^2+^ transport pathway, suggesting that MFF on MOM might participate in VDAC-mediated Ca^2+^ entry. To test this possibility, an IP analysis was performed using anti-MFF and anti-VDAC1 antibodies in untreated DNs. A Western blot analysis showed that the anti-MFF antibody co-precipitated VDAC1 with MFF (Figure 4c, Appendix A). DRP1 was also co-precipitated under this condition (Figure 4c, Appendix A), suggesting that the anti-MFF antibody captured DRP1-mediated fission machinery and Ca^2+^ transport machinery, both of which might share MFF. Alternatively, VDAC1 is an essential component of Ca^2+^ transport machinery, but not of DRP1-mediated fission machinery. To exclusively capture Ca^2+^ transport machinery but not DRP1-mediated fission machinery, IP was performed using an anti-VDAC1 antibody. A Western blot analysis showed that MFF, and not DRP1, was co-precipitated with VDAC1 using the anti-VDAC1 antibody (Figure 4c, Appendix A), suggesting that the anti-VDAC1 antibody might capture the Ca^2+^ transport machinery containing VDAC1 and MFF, but it does not capture the DRP1-mediated fission machinery lacking VDAC1. Consistent with the reduction in mitochondrial Ca^2+^ levels, mitochondrial ROS levels were recovered in the MFF-siR group that was treated by Xest-C (Figure 4d). An immunofluorescence co-staining analysis using SEC61 translocon subunit beta (SEC61B), a marker of ER, and Tom20, a marker of mitochondria, suggested increased contact between ER and mitochondria in the MFF-siR group (Appendix A). These results suggest that MFF is involved in negatively modulating Ca^2+^ influx as a component of Ca^2+^ transport machinery, and therefore, MFF insufficiency may contribute to mitochondrial Ca^2+^ accumulation, triggering ROS overproduction.

### 3.5. Effect of Folic Acid on Developmental Defects of Dopaminergic Neurons with MFF Insufficiency

We hypothesized that the reduced mitochondrial amount and impaired neurite development in the DNs with MFF insufficiency were associated with mitochondrial Ca^2+^-triggered ROS overproduction. To test this hypothesis, FA was used in this study, because it affects ROS scavenging and mitochondrial activation via one-carbon metabolism [42,43,44]. FA supplementation did not alter the mitochondrial morphology and Ca^2+^ accumulation in the MFF-siR group (Appendix A). However, FA supplementation recovered mitochondrial ROS levels in the MFF-siR group (Figure 5a). Transcription of the endogenous antioxidant enzymes SOD1 and SOD2 is sensitive to ROS levels. Both SOD1 and SOD2 transcripts were upregulated in the MFF-siR group and were downregulated by FA supplementation, in correlation with ROS levels (Figure 5b). FA supplementation improved the mitochondrial amount and distribution in the MFF-siR group (Figure 5c).

These improvements correlated with the alteration of both mRNA and protein levels of PGC-1α, a master regulator of mitochondrial biogenesis (Figure 6a,b). This was also supported by the expression levels of mitochondrial transcription factor A (TFAM), a downstream target of PGC-1α, in the MFF-siR group before and after FA supplementation (Figure 6c). FA supplementation did not alter MMP but increased the ATP levels (Figure 6d).

Finally, both neurite development and KCl-induced dopamine secretion were recovered by FA supplementation in MFF-siR group (Figure 7a,b). Thus, FA might slow the degenerative process of DNs due to MFF insufficiency by accelerating ROS scavenging and mitochondrial biogenesis.

## 4. Discussion

The current study aimed to elucidate the neuropathological mechanisms of EMPF2 that is caused by loss-of-function mutations in MFF. In addition to mitochondrial elongation, Ca^2+^ that was actively transported from the ER via IP3R and mtCU was observed to accumulate in the mitochondria of DNs with MFF insufficiency. An IP analysis using anti-MFF and anti-VDAC1 antibodies suggested that MFF might be included in the Ca^2+^ transport machinery that was associated with VDAC1. Mitochondrial Ca^2+^ accumulation that was caused by MFF insufficiency triggered ROS production, leading to impairments in neurite development and PGC-1α-mediated mitochondrial biogenesis. FA supplementation reversed these defects by activating ROS scavenging and PGC-1α-mediated mitochondrial biogenesis. MFF might negatively modulate mitochondrial Ca^2+^ levels in the Ca^2+^ transport pathway and MFF insufficiency might cause mitochondrial Ca^2+^ accumulation that is associated with oxidative stress and impaired mitochondrial biogenesis, leading to the characteristic neurodevelopmental defects that are observed in EMPF2. These defects could be ameliorated by FA.

Mitochondrial Ca^2+^ levels are regulated by the balance between their influx and efflux, and excessive Ca^2+^ influx and/or restricted Ca^2+^ efflux leads to mitochondrial Ca^2+^ accumulation [37,38]. The role of MFF in these Ca^2+^ transport pathways is not fully understood. Previously, mitochondrial Ca^2+^ accumulation has also been reported in mouse cortical pyramidal neurons, in which MFF was suppressed by siRNA in vivo, but the exact mechanisms are unclear [23]. Current data suggest that in DNs with MFF insufficiency, elevated mitochondrial Ca^2+^ levels are predominantly due to active Ca^2+^ influx through IP3R in the ER membrane and mtCU in the MIM. This Ca^2+^ transport pathway is associated with a local structure called the mitochondria-associated membrane (MAM), where the ER membrane and MOM are closely associated [40,45]. Although the molecules that are involved in this pathway have not been completely determined, IP3R on the ER membrane and VDACs on the MOM have been shown to physically interact via the chaperone glucose regulatory protein 75 (grp75) [46]. MFF is tethered by its C-tail anchor on the MOM and the remaining N-terminal region is in the cytosol [17,18]. In the present study, MFF was co-immunoprecipitated with VDAC1, the most common isoform of VDACs in mammals [47]. Overexpressed MFF has been shown to interact with VDAC1 to negatively regulate VDAC1-mediated MOM permeability in non-small cell lung cancer [48]. Although the precise molecular mechanisms remain unresolved, MFF might be a functional component of the Ca^2+^ transport pathway in MAM, participating in the regulation of mitochondrial Ca^2+^ levels as a negative modulator during neurodevelopment.

Elevated physiological levels of mitochondrial Ca^2+^ activate the TCA cycle, accelerating electron transfer in the respiratory chain complexes and causing an increase in membrane potential [49]. Despite increased electron leakage and ROS generation in this process, physiological ROS levels are maintained by activating endogenous antioxidant systems [3,50]. However, continuous Ca^2+^ overload can lead to an overproduction of ROS beyond physiological thresholds and further disrupt mitochondrial function, including persistent opening of the permeability transition pore and depolarization of the MIM, and ultimately cell death [3,24,25]. Mitochondrial Ca^2+^ accumulation in DNs with MFF insufficiency triggered ROS generation, correlating with the upregulation of SOD1 and SOD2, two major endogenous anti-ROS enzymes. However, as shown previously [19,20,21,22,23], DNs with MFF insufficiency survived and even maintained MMP and ATP levels, suggesting that mitochondrial Ca^2+^ and ROS levels were not lethal. Given that MFF-siRNA did not affect mitochondrial Ca^2+^ efflux in DNs, Ca^2+^ could possibly be actively extruded from the mitochondria to avoid destructive Ca^2+^ overload. Ca^2+^ efflux from the mitochondria is most dependent on the Na^+^/Ca^2+^, Li^+^ exchanger that is located in the MIM, which leads to MIM depolarization due to the electrogenic 3 Na^+^:1 Ca^2+^ exchange [51,52]. Therefore, MFF insufficiency could activate mitochondrial Ca^2+^ influx that could elevate MMP and stimulate ROS generation, but also accelerate Ca^2+^ efflux, which might result in MIM depolarization before coupling with ATP synthesis.

The mitochondrial Ca^2+^-triggered ROS production in MFF insufficiency might contribute to the inhibition of mitochondrial biogenesis in DNs. In mammals, ROS levels are closely associated with mitochondrial biogenesis, which is positively or negatively regulated depending on the type of stress that is induced [53,54]. PGC-1α is a master regulator for mitochondrial biogenesis [55]. Acute or moderate oxidative stress has been shown to adaptively induce PGC-1α-mediated mitochondrial biogenesis, while chronic or extensive stress can induce the opposite result [54,55,56,57,58,59,60]. The downregulation of PGC-1α and reduced number of mitochondria that are observed in DNs with MFF insufficiency suggest potential mechanisms of the adverse effects of ROS that are triggered by mitochondrial Ca^2+^ accumulation. This possibility is supported by data showing that FA supplementation suppressed ROS levels, upregulated PGC-1α, and restored mitochondrial amount and neurite development in the MFF-siRNA group, without affecting mitochondrial morphology and Ca^2+^ levels. The health benefits of FA supplementation have been recognized, and their effects have been shown to be mediated through multiple mechanisms, including one-carbon metabolism donating methyl-groups and free radical scavenging [42,43]. The mechanisms of FA action on DNs with MFF insufficiency have not been fully determined in the current study. However, the pathways and molecules regulating mitochondrial ROS levels and biogenesis might be potential therapeutic targets for the neuropsychiatric symptoms of EMPF2.

DRP1 and MFF, which function cooperatively as critical components of the mitochondrial fission machinery, may also have mutually independent functions during neurodevelopment. Loss-of-function mutations of either DRP1 and MFF cause different neurodevelopmental defects—EMPF1 and EMPF2, respectively. However, mitochondrial over-elongation due to impaired fission is a common symptom in both the defects [6,7,8,9,10,11,12,13,19,20,21,22]. Non-overlapping mechanisms could also be present between EMPF1 and EMPF2 neuropathologies, thus underscoring differential roles of DRP1 and MFF in neurodevelopment. This possibility is supported by the observation that DRP1 downregulation did not clearly affect mitochondrial Ca^2+^ levels in DNs. DRP1 is recruited from the cytosol to the mitochondria targeting MFF, MID49, or MID51 on the MOM in response to mitochondrial fission signals, indicating that the primary role of DRP1 on mitochondria is in executing fission [1,2]. Thus, the complete loss of DRP1-dependent mitochondrial fission might be the predominant cause of EMPF1 neuropathology that leads to mitochondrial aggregation and prevents the distribution of active mitochondria into neurites, which can disrupt neurite elongation and branching [14]. In contrast, MFF is constitutively expressed on the MOM, suggesting that MFF may contribute to Ca^2+^ transport together with VDAC1 under conditions that are not involved in DRP1-mediated mitochondrial fission. In EMPF2 carrying intact DRP1, MID49, and MID51, mitochondrial Ca^2+^ dysregulation may be further highlighted in neuropathology, rather than mitochondrial morphological dysregulation.

This study has several limitations. First, the siRNA-based gene knockdown technique did not completely block the MFF expression. To exclude the effects of residual MFF function, SHEDs that are derived from patients who are affected by EMPF2 can be used for future analysis, even though EMPF2 is a rare mitochondrial disease. Second, the molecular mechanisms of the MFF-mediated modulation of Ca^2+^ transport are not fully understood. Further studies on Ca^2+^ transport machinery are required, including determination of all components, the functional or spatial segregation from the DRP1-mediated fission machinery, and the effects of overexpression of MFF. Third, the comprehensive pathologic mechanisms of the central nerve system that are associated with the neuropsychiatric manifestations of EMPF2 have not yet been fully elucidated. To accomplish this, in vivo models are required. We generated conditional MFF knockout mice using the Cre/loxp system. Using this mouse model, we reported that hepatocyte-specific MFF-deficient mice exhibit a nonalcoholic fatty liver disease-like phenotype [61]. The analysis of mice with disrupted MFF in a neural lineage-specific manner is an ongoing project. Fourth, the molecular mechanisms and pathways of the effects of FA on MFF-insufficient DNs must be elucidated to develop therapeutic strategies for EMPF2.

In conclusion, the current study shows that MFF might contribute to the negative modulation of Ca^2+^ transport from the ER to mitochondria, apart from DRP1-mediated mitochondrial fission. MFF insufficiency caused mitochondrial Ca^2+^ accumulation, triggering excessive ROS production and preventing mitochondrial biogenesis during neurodevelopment, which may participate in the neuropathological mechanisms of EMPF2. Dysregulation of ROS levels and mitochondrial biogenesis is a potential therapeutic target of EMPF2.

## Figures and Tables

**Figure 1 antioxidants-11-01361-f001:**
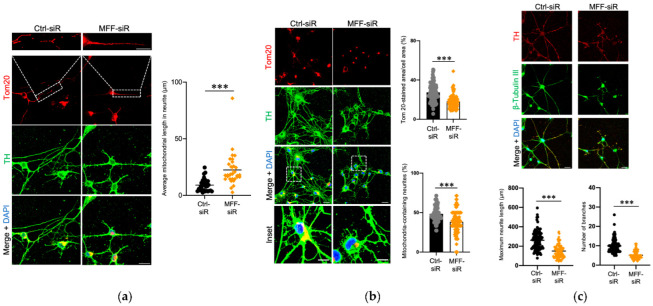
MFF knockdown induced mitochondrial elongation and developmental defects in the neurites of DNs. Stem cells from human exfoliated deciduous teeth were transfected with negative control- (Ctrl-siR) and MFF-siRNA (MFF-siR) and differentiated into DNs. (**a**,**b**) The DNs were stained with anti-Tom20 and anti-TH antibodies and counterstained with DAPI. (**a**) Mitochondrial length of neurites was measured for 10 mitochondria in each group. Scale bar = 20 μm. The boxed regions on the Tom20-stained images are shown at a greater magnification in the upper panels. Scale bar = 10 μm. The mean ± SEM was taken from three independent experiments. *** *p* < 0.001. (**b**) Tom20-stained area per cell area and the percentage of mitochondria-containing neurites were measured for 30 cells in each group. Scale bars = 25 μm. The boxed regions on the merged images are shown at a greater magnification in the lower panels. Scale bars = 10 μm. The mean ± SEM was taken from three independent experiments. *** *p* < 0.001. (**c**) The DNs were stained with anti-β-tubulin III, anti-TH antibodies, and DAPI. Scale bars = 25 μm. Maximum neurite length and number of branches per cell were examined. The mean ± SEM was taken from three independent experiments. *** *p* < 0.001.

**Figure 2 antioxidants-11-01361-f002:**
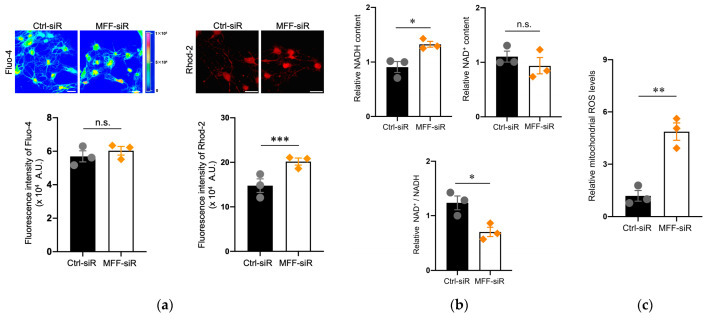
MFF insufficiency induced mitochondrial Ca^2+^ and ROS accumulation in DNs. (**a**) DNs stained with Fluo-4 AM (to detect cellular Ca^2+^) and Rhod-2 AM (to detect mitochondrial Ca^2+^) were observed by confocal microscopy. Scale bars = 25 μm. Fluorescence intensity of Fluo-4 AM and Rhod-2 AM were measured using a plate reader. The mean ± SEM was taken from three independent experiments. n.s., not significant, *** *p* < 0.001. (**b**) Relative levels of intracellular NADH, NAD^+^, and NAD^+^/NADH in each case. The mean ± SEM was taken from three independent experiments. n.s., not significant, * *p* < 0.05. (**c**) Mitochondrial ROS levels were measured using flow cytometry. The mean ± SEM was taken from three independent experiments. ** *p* < 0.01.

**Figure 3 antioxidants-11-01361-f003:**
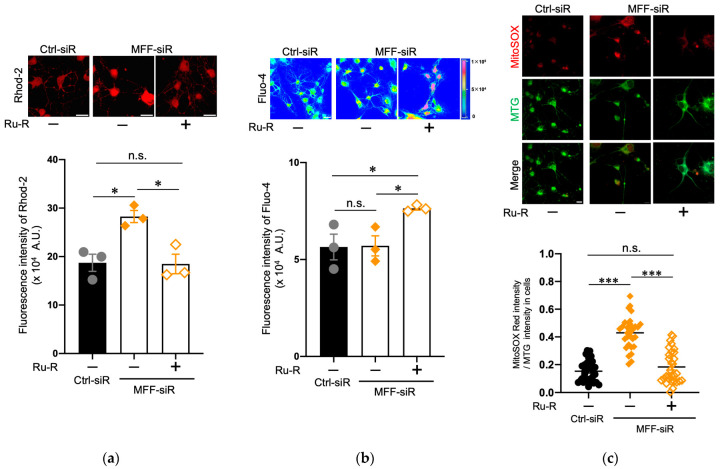
Mitochondrial calcium uniporter channel inhibitor Ru-R suppressed mitochondrial Ca^2+^ and ROS accumulation in DNs with MFF insufficiency. Stem cells from human exfoliated deciduous teeth were differentiated into DNs in the absence or presence of Ru-R. (**a**) DNs stained with Rhod-2 AM were observed by confocal microscopy. Scale bars = 25 μm. Fluorescence intensity of Rhod-2 AM was measured using a plate reader. The mean ± SEM was taken from three independent experiments. n.s., not significant, * *p* < 0.05. (**b**) DNs stained with Fluo-4 AM were observed by confocal microscopy. Scale bar = 25 μm. Fluorescence intensity of Fluo-4 AM was measured using a plate reader. The mean ± SEM was taken from three independent experiments. n.s., not significant, * *p* < 0.05. (**c**) DNs stained with MitoSOX Red and MTG were observed by confocal microscopy. Scale bar = 25 μm. To measure the ROS level per mitochondrion, the fluorescence intensity of MitoSOX Red was divided by that of MTG. The mean ± SEM was taken from three independent experiments. n.s., not significant, *** *p* < 0.001.

**Figure 4 antioxidants-11-01361-f004:**
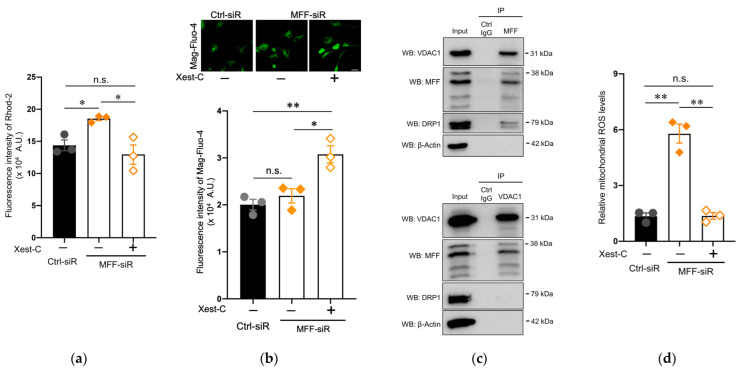
IP3R inhibitor Xest-C suppressed mitochondrial Ca^2+^ and ROS accumulation in DNs with MFF insufficiency. Stem cells from human exfoliated deciduous teeth were differentiated into DNs in the absence or presence of Xest-C. (**a**) DNs were stained with Rhod-2 AM. The fluorescence intensity of Rhod-2 AM was measured using a plate reader. The mean ± SEM was taken from three independent experiments. n.s., not significant, * *p* < 0.05. (**b**) DNs stained with Mag-Fluo-4 AM were observed by confocal microscopy. Scale bar = 25 μm. The fluorescence intensity of Mag-Fluo-4 AM was measured using a plate reader. The mean ± SEM was taken from three independent experiments. n.s., not significant, * *p* < 0.05, ** *p* < 0.01. (**c**) IP was performed using an anti-MFF (upper panel) and anti-VDAC1 (lower panel) antibodies in untreated DNs. Immunoprecipitants were detected by Western blotting (WB) using the indicated antibodies. (**d**) Mitochondrial ROS levels were measured using flow cytometry. The mean ± SEM was taken from three independent experiments. n.s., not significant. ** *p* < 0.01.

**Figure 5 antioxidants-11-01361-f005:**
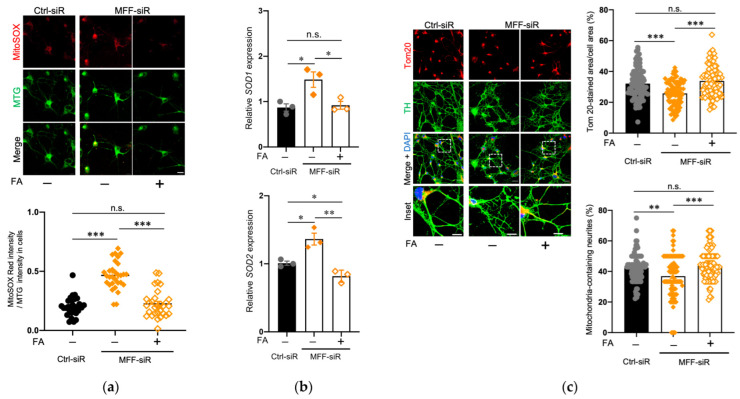
FA supplementation improved mitochondrial ROS levels and mitochondrial amount and distribution in DNs with MFF insufficiency. Stem cells from human exfoliated deciduous teeth were differentiated into DNs either in the absence or presence of FA. (**a**) DNs were stained with MitoSOX Red and MTG. Scale bar = 20 μm. To measure the ROS level per mitochondrion, the fluorescence intensity of MitoSOX Red was divided by that of MTG. The mean ± SEM was taken from three independent experiments. n.s., not significant, *** *p* < 0.001. (**b**) *Superoxide dismutase 1* and *2* (SOD1 and SOD2) mRNA expression in DNs was measured using RT-qPCR. The mean ± SEM was taken from three independent experiments. n.s., not significant, * *p* < 0.05, ** *p* < 0.01. (**c**) DNs were stained with anti-Tom20 and anti-TH antibodies and counterstained with DAPI. Scale bars = 20 μm. Boxed regions on the merged images are shown at a greater magnification in the lower panels. Scale bars = 10 μm. Tom20-stained area per cell area and the percentage of mitochondria-containing neurites were measured. The mean ± SEM was taken from three independent experiments. n.s., not significant, ** *p* < 0.01, *** *p* < 0.001.

**Figure 6 antioxidants-11-01361-f006:**
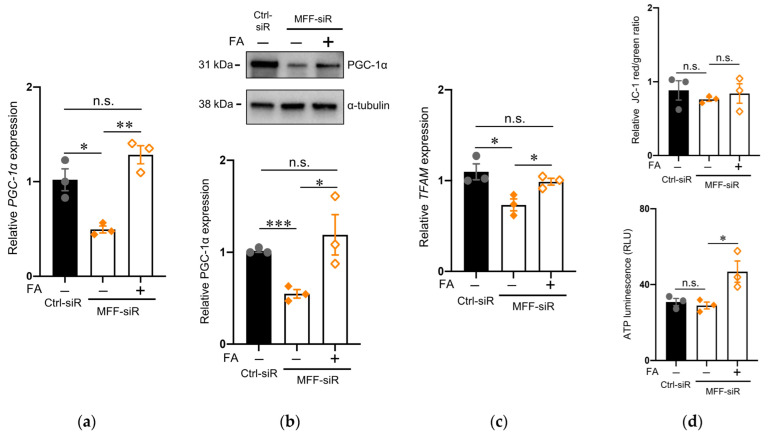
FA supplementation improved mitochondrial biogenesis in DNs with MFF insufficiency. (**a**,**c**) PGC-1α and TFAM mRNA expression in DNs was measured using RT-qPCR. The mean ± SEM was taken from three independent experiments. n.s., not significant, * *p* < 0.05, ** *p* < 0.01. (**b**) PGC-1α protein expression levels were measured by Western blotting. The mean ± SEM was taken from three independent experiments. n.s., not significant, * *p* < 0.05, *** *p* < 0.001. (**d**) Mitochondrial membrane potential was measured with JC-1. The ratio of JC-1 red to green was calculated. ATP levels were measured using luminescence assays. ATP luminescence signals were divided by the number of cells. The mean ± SEM was taken from three independent experiments. n.s., not significant, * *p* < 0.05.

**Figure 7 antioxidants-11-01361-f007:**
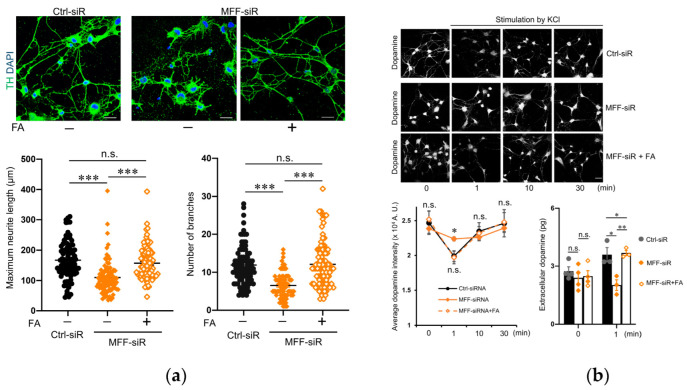
FA supplementation improved developmental defects in DNs with MFF insufficiency. (**a**) DNs were stained with anti-TH antibodies and DAPI. Scale bars = 25 μm. Maximum neurite length and number of branches per cell are shown. The mean ± SEM was taken from three independent experiments. n.s., not significant, *** *p* < 0.001. (**b**) DNs were stimulated with 50 mM of KCl for the indicated times and subsequently stained with anti-dopamine antibodies. Scale bar = 25 μm. Dopamine staining intensity per cell area was measured for 30 cells of each case. The mean ± SEM was taken from three independent experiments. Extracellular dopamine levels were measured under basal conditions and a 50 mM-KCl stimulated condition. The mean ± SEM was taken from three independent experiments. n.s., not significant, * *p* < 0.05, ** *p* < 0.01.

**Table 1 antioxidants-11-01361-t001:** Primers sequences.

Gene	Accession Number		Sequence (5′-3′)
18S	X03205.1	Forward	CGGCTACCACATCCAAGGAA
		Reverse	GCTGGAATTACCGCGGCT
SOD1	NM_000454.5	Forward	GGTGGGCCAAAGGATGAAGAG
		Reverse	CCACAAGCCAAACGACTTCC
SOD2	NM_000636.4	Forward	AAACCTCAGCCCTAACGGTG
		Reverse	GCCTGTTGTTCCTTGCAGTG
*PGC-1α*	NM_001330751.2	Forward	GGCAGAAGGCAATTGAAGAG
		Reverse	TCAAAACGGTCCCTCAGTTC
*TFAM*	NM_001270782.2	Forward	GATGCTTATAGGGCGGAGTGG
		Reverse	GCTGAACGAGGTCTTTTTGGT

## Data Availability

Data are contained within the article and Appendix A.

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
