# Peer review of "Mitochondrial Calcium-Triggered Oxidative Stress and Developmental Defects in Dopaminergic Neurons Differentiated from Deciduous Teeth-Derived Dental Pulp Stem Cells with MFF Insufficiency"

_antioxidants, 2022, doi:10.3390/antiox11071361_

Round 1
Reviewer 1 Report
Major point:
1) Rhod-2 dye does not seem to specifically delineate the DN mitochondrial compartment in Figures 2A, 3A when compared to Tom20 (Figures 1A,B, 5C, S1A, S8B), MTG (Figure 3C), or MitoSOX (Figure 3C, 5A) as the whole cell interior is stained. This fact therefore represents a major issue especially when comparing Rhod-2 signal to the more general Fluo-4 Ca2+ sensor. Please either improve the mitochondrial Ca2+ imaging technique to yield more mitochondria-resembling readout or discuss this limitation in the text.
2) Despite the authors claim that MFF depletion has no effect on apoptosis and cell viability (Figure S4), there seems to be a considerable reduction in the amount of DAPI-stained cells in the MFF-siR condition when compared to Ctrl-siR (Figure S4B). Please resolve or explain this dichotomy in the text.
Minor points:
1) Please replace "DNs" with "dopaminergic neurons" (lines 86, 145, 274, 312, 346, 377, 424).
2) Please change "37°C" to "37 °C" (line 96).
3) Please provide catalog number for DMEM mentioned in "After 24 h, the cells were cultured in the first step medium, Dulbecco’s Modified Eagle’s Medium (DMEM, Nacalai Tesque), supplemented with 20 ng/mL epidermal growth factor (PeproTech, NJ, USA), 20 ng/mL basic fibroblast growth factor (Peprotech), and 1 % N2 supplement (Thermo Fisher Scientific, MA, USA) for two days, at 37 °C, in an incubator with 5 % CO2" (line 100).
4) Please change "days, at 37 °C, in" to "days at 37 °C in" (line 104).
5) Please provide catalog number for Alexa Fluor-conjugated secondary antibodies mentioned in "The cells were subsequently incubated with Alexa Fluor-conjugated secondary antibodies (Thermo Fisher Scientific) for 1 h at room temperature in the dark" (line 136).
6) Please replace "ER" with "endoplasmic reticulum" (line 207).
7) Please change "Endoplasmic Reticulum" to "endoplasmic reticulum" (line 209).
8) It is not clear what the authors mean by "control- and patient-derived case" in "To measure intracellular dopamine levels, the images of dopamine and TH (in the cell area) staining were acquired and 30 cells of each control- and patient-derived case were analyzed with the Multi Wavelengths Cell Scoring module in MetaMorph software version 7.8. The dopamine signal intensity was divided by the TH-stained surface area" (line 225)?
9) Please replace "MitoTracker Green FM" with "MitoTracker Green FM (MTG) (line 248)", "MitoTracker Green FM" with "MTG" (lines 249, 251), and "MitoTracker Green (MTG)" with "MTG" (line 372, 441).
10) Please change "red/green" to "red to green" (line 461).
11) Please replace "shorter" with "longer" (line 287).
12) Please either change "Merge" to "Merge + DAPI" or provide the "DAPI channel" in Figures 1A,B,C.
13) The resolution of the imaging panels in Figure 1C is poor. Please increase size and/or pixel density so that cellular morphologies can be clearly distinguished.
14) The TH signal is hardly visible in Figure 1C. Could the authors please increase its intensity and replot the respective figure panels?
15) Please change "Mitochondrial fission factor (MFF)" to "MMF" (lines 297, 335).
16) Please remove bold formatting from "(" in "(MFF)" (line 297).
17) Please replace "dopaminergic neurons (DNs)" with "DNs" (lines 298, 336, 364, 410, 438).
18) Please remove bold formatting from "," in "(a, b)" (line 300).
19) Please change "4ËŠ,6-diamidino-2-phenylindole dihydrochloride (DAPI)" to "DAPI" (line 301).
20) Please change "bars" to "bar" (lines 303, 367, 370, 373, 416, 441).
21) Please change "standard error of the mean (SEM)" to "SEM" (lines 304, 339, 368, 414, 442, 462, 477).
22) Please replace "area/cell" with "area per cell" (lines 305, 449).
23) Please replace "bar" with "bars" (lines 306, 309, 448, 476).
24) The resolution of the imaging panels in Figure 2A is poor. Please increase size and/or pixel density so that cellular morphologies can be clearly distinguished.
25) Please replace "induce" with "induce apoptotic" (line 325).
26) It is not exactly clear what the authors mean by "elongated by the defects in DRP1" "Next, we examined whether mitochondrial Ca2+ and ROS accumulation also occur in mitochondria that were elongated by the defects in DRP1" (line 326)? What concrete defects in DRP1 are the authors referring to?
27) Please change "occur" to "occurred" (line 327).
28) Please provide reference for "However, as shown in previous reports, mitochondrial ROS levels were elevated by DRP1 inhibition" (line 330).
29) Please provide intensity heat map for the Fluo-4 imaging panels in Figure 2A.
30) Please provide reference for "The mitochondrial calcium uniporter channel (mtCU) is located in the mitochondrial inner membrane (MIM) and is one of the critical regulators of Ca2+ entry into the mitochondrial matrix" (line 347).
31) It is not clear for how long was Ru-R applied in "To elucidate the mechanisms of mitochondrial Ca2+ accumulation caused by MFF insufficiency, mtCU was blocked by its specific reagent, Ru-R" (line 349)?
32) Please provide reference for "To elucidate the mechanisms of mitochondrial Ca2+ accumulation caused by MFF insufficiency, mtCU was blocked by its specific reagent, Ru-R" (line 349).
33) Please define abbreviation for "Ru-R" (line 350).
34) From "Upon Ru-R treatment, the elevated mitochondrial Ca2+ levels in the MFF-siR group were reduced to levels comparable to the untreated Ctrl-siR group (Fig. 3a), while cytosolic Ca2+ levels were increased in the MFF-siR group treated with Ru-R (Fig. 3b)" (line 350) is not clear why were cytosolic Ca2+ levels "increased in the MFF-siR group treated with Ru-R"?
35) The meaning of "suggesting no critical defects of Ca2+ efflux" is not entirely clear in "This treatment could also reduce the mitochondrial Ca2+ levels in the MFF-siR group to levels similar to that of the Ctrl-siR group, suggesting no critical defects of Ca2+ efflux" (line 354) as 4 h Ru-R treatment seems to be tested only to shorten the time for which cells are exposed to this inhibitor but not to directly exclude defects in mitochondrial Ca2+ efflux.
36) Please replace "could also reduce" with "also reduced" (line 355).
37) The resolution of the imaging panels in Figures 3A,B,C is poor. Please increase size and/or pixel density so that cellular morphologies can be clearly distinguished.
38) Please label the respective column of micrographs as "Ctrl-siR" in Figure 3C.
39) Please change "Mito Sox" to "MitoSOX" in Figure 3C.
40) Please replace "reactive oxygen species (ROS)" with "ROS" (lines 364, 410, 437).
41) Please change "mitochondrial fission factor (MFF)" to "MFF" (lines 365, 411, 459, 475).
42) Please provide reference for "The inositol 1,4,5-trisphosphate receptor (IP3R) of the ER membrane is a regulatory component of this pathway" (line 380).
43) Please provide reference for "To clarify the role of the ER in mitochondrial Ca2+ accumulation caused by MFF-insufficiency, we used Xest-C, a specific inhibitor of IP3R" (line 382).
44) Please replace "MFF-insufficiency" with "MFF insufficiency" (line 382).
45) Please change "was clearly" to "was" (line 399).
46) Please replace "lacking in" with "lacking" (line 402).
47) The resolution of the imaging panels in Figure 4B is poor. Please increase size and/or pixel density so that cellular morphologies can be clearly distinguished.
48) Please replace "Xestospongin C (Xest-C)" with "Xest-C" (line 409).
49) Please change "FA" to "folic acid" (line 424).
50) Please replace "The reduced mitochondrial amount and impaired neurite development in the DNs with MFF insufficiency were to be" with "We hypothesized that the reduced mitochondrial amount and impaired neurite development in the DNs with MFF were" (line 425).
51) Please provide reference for "To test this hypothesis, FA was used in this study, because it affects ROS scavenging and mitochondrial activation via one-carbon metabolism" (line 427).
52) Please replace "quantitative reverse transcription polymerase chain reaction" with "RT-qPCR" (lines 445, 461, 476).
53) Please change "4ËŠ,6-diamidino-2-phenylindole dihydrochloride" to "DAPI" (lines 447, 476).
54) The resolution of the imaging panels in Figures 5A,C is poor. Please increase size and/or pixel density so that cellular morphologies can be clearly distinguished.
55) Please change "MitoSox" to "MitoSOX" in Figure 5A.
56) Please either change "Merge" to "Merge + DAPI" or provide the "DAPI channel" in Figure 5C.
57) Please replace "Peroxisome proliferator-activated receptor gamma coactivator 1-alpha (PGC-1α) and mitochondrial transcription factor A (TFAM)" with "PGC-1α and TFAM" (line 459).
58) Please change "JC-1 red/green" to "red to green JC-1" (line 466).
59) Please replace "x 104" with "x 104 A.U." in Figure 7B.
60) Please change "Non-overlapped" to "Non-overlapping" (line 565).
61) Please replace "thus" with something like "thus underscoring" (line 566).
62) Please change "targeting any of" to "targeting" (line 569).
63) Please replace "S.D." with "S.D.," (line 608).
64) Please change "Y.I." to "Y.I.," (line 609).
65) Please replace "S.D" with "S.D." (line 609).
66) Please change "Y.S." to "Y.S.," (line 610).
67) Please replace "Y.S., S.O." with "Y.S., and S.O." (line 611).
68) Please change "Y.H." to "Y.H.," (line 611).
69) Please replace "JP19K10406" with "JP19K10406," (line 614).
70) Please format "a" in "(a)" using bold formatting in the legend to Figures S1, S2, S3, S4, S5, S8.
71) Please replace "Dopaminergic neurons (DNs)" with "DNs" in the legend to Figures S1, S3, S6, S8.
72) Please change "Scale bar = 10 μm" to "Scale bars = 10 μm" in the legend to Figure S1.
73) Please change "2μm" to "2 μm" in the legend to Figure S1.
74) Please format "b" in "(b)" using bold formatting in the legend to Figures S1, S2, S3, S4, S5, S8.
75) Please change "standard error of the mean (SEM)" to "SEM" in the legend to Figures S1, S2, S3 2x, S4, S5 4x, S6, S8.
76) Please format "c" in "(c)" using bold formatting in the legend to Figures S1, S5, S6.
77) Please change "dopaminergic neurons (DNs)" to "DNs" in the legend to Figures S2, S4.
78) Please replace "quantitative reverse transcription-polymerase chain reaction" with "RT-qPCR" in the legend to Figure S2.
79) Please change "quantitative reverse transcription-polymerase chain reaction" to "quantitative reverse transcription-polymerase chain reaction (RT-qPCR)" in the legend To Figure S2.
80) Please replace "MitoTracker Green (MTG)" with "MTG" in the legend to Figure S3.
81) Please change "Scale bars" to "Scale bar" in the legend to Figure S3 2x.
82) Please change "in neurite" to "in neurites" in the legend to Figure S3 2x.
83) Please replace "Scale bar" with "Scale bars" in the legend to Figure S4.
84) Please comment on in the text why Drp1-siR neurons appear thicker and contain more branch points when compared to their Ctrl-siR counterparts in Figure S5B?
85) Please replace "mitochondrial fission factor (MFF)" with "MFF" in the legend to Figure S5.
86) It is not clear whether the sense and antisense sequences represent RNA or DNA in "Dynamin-related protein 1 (DRP1) and mitochondrial fission factor (MFF) protein levels in DRP1-siRNA, sense 5'-GUAAUACUGAGACUUUGUUTT-3' and antisense 5'-AACAAAGUCUCAGUAUUACTT-3', treated cells (DRP-siR) were measured using western blotting" in the legend to Figure S5 as both uracil and thymine bases are both present? Please revise.
87) From "Dynamin-related protein 1 (DRP1) and mitochondrial fission factor (MFF) protein levels in DRP1-siRNA, sense 5'-GUAAUACUGAGACUUUGUUTT-3' and antisense 5'-AACAAAGUCUCAGUAUUACTT-3', treated cells (DRP-siR) were measured using western blotting" in the legend to Figure S5 is also not clear which sequence corresponds to Ctrl-siR and which to DRP1-siR in Figure S5A?
88) Please format "d" in "(d)" using bold formatting in the legend to Figure S5.
89) Please change "MitoSOX red" to "MitoSOX Red" in the legend to Figures S5, S6.
90) Please format "a" and "b" in "(a, b)" using bold formatting in the legend to Figures S6, S7.
91) Please either change "Merge" to "Merge + DAPI" or provide the "DAPI channel" in Figure S8B.
92) Please change "Scale bar" to "Scale bars" in the legend to Figure S8.
93) Please change "Scale bars" to "Scale bar" in the legend to Figure S8.
94) Please replace "10 mitochondrial" with "10 mitochondria" in the legend to Figure S8.
Reviewer 2 Report
This manuscript used stem cells from human exfoliated deciduous teeth (SHEDs) and their differentiation potential into dopaminergic neurons (DNs) as a disease model of EMPF2. The authors applied this model to clarify the effects of Mff knock down and to elucidate the neuropathological mechanisms of EMPF2. The results demonstrated that treating with Mff-targeting small interfering RNA caused impaired neurite outgrowth and reduced mitochondrial signals in neurites harboring elongated mitochondria. In addition, it caused mitochondrial Ca2+ and ROS accumulation The accumulation potentially depends on the inositol 1,4,5-trisphosphate receptor. Furthermore, downregulated peroxisome proliferator-activated receptor gamma co-activator-1 alpha (PGC-1α). Mff was co-immunoprecipitated with voltage-dependent anion channel 1. Folic acid supplementation ameliorated the accumulation of ROS levels, rescued PGC-1α-mediated mitochondrial biogenesis, and neurite outgrowth in MFF-depleted DNs, without affecting their mitochondrial morphology or Ca2+ levels. The manuscript contains lots of finding related to manipulating Mff in DNs. It will be interesting to readers from related research fields.
Major concerns
l The results of this manuscript are associated with Mff depletion. However, the cause-effect remain obscure. It is the reviewer’s comments that the authors need to address the issue more carefully to describe the finding precisely.
l The manuscript manipulating Mff expression level with RNAi. However, if will be interesting to see the effects of overexpressed Mff in the current model.
l Even mitochondria morphology remained elongated in Mff depletion, variety changes of mitochondria related activity parameters were detected. It will be interesting to know whether ER-mitochondria contacts were affected under Mff depletion condition.
Round 2
Reviewer 1 Report
Sun and Dong et al. have scrutinized the impact of the mitochondrial fission receptor Mff, residing on the outer mitochondrial membrane (OMM), on redox and developmental homeostases in dopamine-producing neurons obtained from deciduous teeth-derived dental pulp stem cells (DPSCs) by in vitro differentiation. By deploying a profound toolkit of molecular, biochemical, and cell biological approaches, the authors have come up with a thorough picture of the wide-ranging effects that Mff depletion has in neurons differentiated from DPSCs. First, they have revealed that the underlying mechanism involves inositol 1,4,5-trisphosphate receptor (IP3R)-dependent mitochondrial Ca2+ influx from the endoplasmic reticulum, which resulted in increased generation of redox equivalents originating in the tricarboxylic acid cycle and concomitantly elevated production of mitochondrial superoxide. Accordingly, these functional changes led to mitochondrial elongation and neuronal morphology aberrations. Consistent with the proposed hypothesis, some of these phenotypes were recapitulated following Drp1 depletion. Moreover, Mff-mediated alternations were causally linked to peroxisome proliferator-activated receptor-gamma co-activator-1 alpha (PGC-1α) downregulation. Further biochemical analyses suggested that this key component of the mitochondrial machinery modulates mitochondrial Ca2+ influx at the OMM through its interaction with voltage dependent anion channel 1 (VDAC1). Importantly, Sun and Dong et al. highlighted the promising neuroprotective utility of folic acid (FA) treatment in DPSC-derived neurons demonstrated by its capacity to restore cellular health as well as the ability of DPSC-derived neurons to secrete dopamine. The question of how FA elicits its therapeutic benefit is nevertheless left unanswered. Overall, the study, which uncovers previously unappreciated role of the mitochondrial fission machinery during neurogenesis, is sound and potentially identifies novel targets for future use in neurotherapy.
1) Please replace "two" with "2" (line 104).
2) Please change "days" to "d" (lines 104, 110).
3) Please replace "37 °C, in" with "37 °C in" (line 104).
4) Please change "five" to "5" (line 109).
5) Please change "Sigma Aldrich" to "Sigma-Aldrich" (line 117).
6) Please replace "temperature" with "temperature," (line 129).
7) Please provide city name for the headquarters of "Dojindo" mentioned in "After staining with secondary antibodies, the nuclei were counterstained with 1 μg/mL of 4′,6-diamidino-2-phenylindole dihydrochloride (DAPI; Dojindo, Japan) in PBS for 5 min at room temperature" (line 139).
8) Please change "buffer, 62.5 mM Tris-HCl buffer" to "buffer containing 62.5 mM Tris-HCl" (line 169).
9) Please replace "at 95 °C for 5 min" with "for 5 min at 95 °C" (line 170).
10) Please replace "BD Biosciences ," with "BD Biosciences," (line 176).
11) Please replace "pro" with "Pro" (line 183).
12) Please change "#X090302," to "#X090302;" (line 193).
13) Please replace "#17090-1-193 AP," with "#17090-1-193 AP;" (line 193).
14) Please change "staining" with "staining with" (line 216).
15) Please change "microscopy, as" to "microscopy as" (line 242).
16) Please provide manufacturer for the "FACSCalibur instrument" mentioned in "The fluorescence signal of 10,000 cells was measured using a FACSCalibur instrument" (line 245).
17) Please replace "Bioscience" with "Biosciences" (line 248).
18) Please change "(Ibidi, Munich, Germany), and subsequently, incubated" to "(Ibidi, Munich, Germany) and subsequently incubated" (line 250).
19) An intelligible set of characters appears when mouse cursor is hovered over both panels in Figure 2a. Please disable this feature.
20) Please replace "reactive oxygen species (ROS)" with "ROS" (line 337).
21) Please change "ruthenium red (Ru-R)" to "Ru-R" (line 368).
22) Please replace "Inositol 1,4,5-trisphosphate receptor" with "IP3R" (line 412).
23) Please change "Immunoprecipitation (IP)" to "IP" (line 419).
24) Please replace "Folic acid (FA)" with "FA" (lines 438, 471).
25) Please change "technique used in this study" to "technique" (line 575).
26) Please move "Immunofluorescence co-staining analysis using SEC61 translocon subunit beta (SEC61B), a marker of ER, and Tom20, a marker of mitochondria, suggested increased contact between ER and mitochondria (Suppl. Fig. S9)" (line 580) into the Results section.
27) Please change "mitochondria" to "mitochondria in the MFF-siR group" (line 582).
28) Please replace "supplemental" with "Supplemental" (line 604 2x).
29) Please change "T.A.K" to "T.A.K." (line 610).
30) An intelligible set of characters appears when mouse cursor is hovered over all panels in Figure S1a. Please disable this feature.
31) Please change "#qHsaCED0001111," to "#qHsaCED0001111;" in the legend to Figure S2.
32) An intelligible set of characters appears when mouse cursor is hovered over all panels in Figure S3a. Please disable this feature.
33) An intelligible set of characters appears when mouse cursor is hovered over the MitoSOX Ctrl-siR panel in Figure S3b. Please disable this feature.
34) Please replace "measured, and" with "measured and" in the legend to Figure S3.
35) An intelligible set of characters appears when mouse cursor is hovered over all panels in Figure S4b. Please disable this feature.
36) An intelligible set of characters appears when mouse cursor is hovered over all panels in Figure S5b. Please disable this feature.
37) Please shorten "ruthenium red (Ru-R)" to "Ru-R" in the legend to Figure S6.
38) Please change "Immunoprecipitation (IP)" to "IP" in the legend to Figure S7.
39) An intelligible set of characters appears when mouse cursor is hovered over all panels in Figure S8b. Please disable this feature.
40) An intelligible set of characters appears when mouse cursor is hovered over all the left panels in Figure S9. Please disable this feature.
